# Enzymatic Management of pH in White Wines

**DOI:** 10.3390/molecules26092730

**Published:** 2021-05-06

**Authors:** Andreea Botezatu, Carlos Elizondo, Martha Bajec, Rhonda Miller

**Affiliations:** 1Horticulture Department, Texas A&M University, College Station, TX 77843, USA; elizondoca36@tamu.edu; 2Bajec Senseworks Consulting, Hamilton, ON L9A 1L5, Canada; martha.bajec@gmail.com; 3Animal Science Department, Texas A&M University, College Station, TX 77843, USA; rmiller@exchange.tamu.edu

**Keywords:** wine, wine pH, glucose oxidase, catalase, hot climate, wine acidity

## Abstract

This paper investigates the potential of the enzymatic management of high pH in white juice and wine using a combination of enzymes-glucose oxidase coupled with catalase. Catazyme^®^ 25 L, a commercially available blend of the two enzymes, was added at different doses (0.2 g/L, 0.6 g/L, and 1g/L) to white grape juice and various parameters (glucose, gluconic acid, pH) were monitored over 24 h of treatment. Treated wines were fermented to dryness without any difficulty and the wines were chemically and sensorially evaluated. At the highest dose (1 g/L), pH was reduced from 3.9 to 3.2, with 20.5 g of gluconic acid produced, while at the lowest dose (0.2 g/L), pH decreased from 4.0 to 3.5 and 8.8 g of gluconic acid was produced. Flash profiling indicated that treated wines were lighter in color than the control and were described using terms such as floral, fruit, citrus, and sour while the control wine was described as being fermented, medicinal, pungent, and oxidized. In conclusion, glucose oxidase coupled with catalase was shown to be effective at significantly reducing juice and wine pH in a short amount of time and with a positive impact on the organoleptic profiles of the treated wines.

## 1. Introduction

There is a sense of urgency within the Texas wine industry as well as in other warm-climate wine regions to solve the problem of high pH, high alcohol wines that are produced in these areas. High pH wines are problematic as they can often be microbiologically unstable, have issues with color stability, and result in organoleptically unbalanced wines [1]. Time of harvest is one way of controlling pH and acidity levels, however, due to the characteristics of the warm growing climates (hot days followed by warm nights without cooling), ripening processes evolve differently, with acidity dropping dramatically before full aroma and phenolic maturity is developed. One potential method to improve wine quality and reduce costs is by increasing acidity and decreasing pH via enzymatic transformation. The enzyme glucose oxidase (GOX), widely used throughout the food and pharmaceutical industry, has been previously investigated for its potential to reduce glucose levels in wines, thus lowering alcohol potential [2,3,4,5,6,7] (GOX is not specifically approved for use in wine in the U.S. yet, but it is approved for use in beer and other food products and is generally regarded as safe (GRAS)-by the Food and Drug Administration. As far as we know, GOX has not been added to the approved wine additive list by the OIV). GOX catalyzes the reaction of β-d-glucose into d-glucono-δ-lactone. Hydrogen peroxide (H_2_O_2)_ is initially formed from the reaction. In the second reaction, d-glucono-δ-lactone is converted into gluconic acid [7,8]. To mitigate the production of H_2_O_2_, the enzyme catalase is routinely used in conjunction with GOX for wine applications [5,8,9]. The conversion of d-glucose into gluconic acid immobilizes its ability to act as a fermentable sugar and increases acidity [5]. The rate of conversion is primarily dependent on oxygen aeration [2]. The reaction mechanism for the GOX-mediated glucose to gluconic acid pathways are shown in Figure 1.

Previous studies investigating the use of the enzyme complex (GOX + catalase) in a wine matrix focused on either using GOX as an oxidation inhibitor in wines with residual sugars [10,11] or in reducing potential alcohol levels in wine by degrading glucose to gluconic acid [2,3,4,5,6,10,11,12,13]. The focus on glucose reduction led to long reaction times (up to 72 h), which in turn led to oxidative issues as well as a high accumulation of gluconic acid (up to 72 g/L [3]) in the final products. High gluconic acid concentrations led, in turn, to wines with extremely low pH, undesirably high titratable acidity, and unbalanced flavor profiles [4]. A limiting factor in previous studies has been low starting wine pH, which inhibits GOX activity, with some authors proposing pre-treatment de-acidification of the experimental wines [2].

Hypothesis: In our study, we propose that GOX in conjunction with catalase can be successfully used to reduce pH and increase titratable acidity (TA) in wines in a short period of time, without negatively affecting the quality of the treated wines. Our focus was not on reducing alcohol levels, although we may see a slight decline in alcohol due to conversion of some glucose to gluconic acid. Since our focus was on high pH wines (pH > 3.8), we also hypothesize that pH will not be a limiting factor in our trials and no de-acidification will be needed, resulting in a fast, easy and inexpensive approach for pH management in hot climate wines.

## 2. Results

### 2.1. Chemical Analysis

After the introduction of Catazyme^®^ 25 L (GOX plus catalase commercial product), Riesling juice was tested every hour (starting at 0 h) for 10 h and then again at 24 h. Four treatments were established prior to the experiment: control (no addition made), 0.2 g/L, 0.6 g/L, and 1 g/L. The focus of this study was to monitor changes in pH, d-glucose, gluconic acid, and color.

pH was tested in triplicate every hour and means were generated from the data collected. Results for pH are listed in Table 1. Initial pH readings for all treatments at 0 h ranged between 3.9 to 4.0. While the control maintained a pH of 4 at 2 h, significant changes (*p* < 0.05) were seen at 2 h. At 2 h, the 0.2 g/L treatment dropped to 3.9, and the 0.6 g/L and 1 g/L treatment dropped to 3.8. As expected, the biggest change happened in the treatments with the highest concentrations of Catazyme^®^ 25 L. The drop in pH continued as each hour passed. At 3 h, 0.6 g/L and 1 g/L observed the biggest decrease with a pH of 3.7, followed by 0.2 g/L with a pH 3.8 (*p* < 0.05). Another significant change occurred at 5 h, where 0.6 g/L and 1 g/L decreased to pH 3.6, and 0.2 g/L to pH 3.7 (*p* < 0.05). At 7 h, there was another significant decrease in pH for 1 g/L with an observed pH 3.5 (*p* < 0.05). No decrease was detected from the previous hour for 0.2 g/L, 0.6 g/L, and the control. At 8 h, a significant drop in pH for treatment 0.2 g/L with a pH 3.6 and 0.6 g/L with a pH 3.5 was observed (*p* < 0.05). The 1 g/L treatment did not change and maintained a pH of 3.5. There was no change in pH at hours 9 and 10. The most significant change in pH occurred at 24 h, where values were different across all treatments (*p* < 0.05). The 1 g/L treatment observed the largest decrease with a pH 3.2, followed by 0.6 g/L at pH 3.3, then 0.2 g/L at pH 3.5. Overall, there was a significant decrease in pH for all treatments outside of the control. The initial and final values for each treatment were as follows: control pH 4.0 to 3.9, treatment A (0.2 g/)L pH 4.0 to 3.5, treatment B (0.6 g/L) pH 3.9 to 3.3, and treatment D (1 g/L) pH 3.9 to 3.2.

As pH drops, it is important to observe the decrease in d-glucose as it is converted to gluconic acid. D-glucose concentrations throughout the experiment can be found in Table 2. D-glucose was tested in triplicate every hour and means were generated from the data collected.

No statistical differences were observed between any of the treatments at any time point with regard to glucose. As a general trend, overall lower values were observed in treated wines as opposed to the control, particularly as time advanced. Glucose concentrations were tested in triplicate every hour and means were generated from data collected.

As d-glucose was enzymatically processed, an increase in gluconic acid was expected.

Gluconic acid concentrations throughout the experiment can be found in Table 3. Gluconic acid concentrations were tested in triplicate every hour and means were generated from the data collected.

The initial gluconic acid concentrations were 627 mg/L for the control, 346 mg/L for the 0.2 g/L treatment, 264 mg/L for 0.6 g/L, and 327 mg/L for the 1 g/L treatment. Gluconic acid production occurred rapidly for all treatments at 1 h. The control stayed the same, while 0.2 g/L went up to 1274 mg/L, 0.6 g/L to 2126 mg/L, and 1 g/L to 4810 mg/L. At 2 h, there was another significant increase in concentration across all treatments: 0.2 g/L to 2259 mg/L, 0.6 g/L to 3717 mg/L, and 1 g/L to 4810 mg/L. The control also saw a slight increase to 708 mg/L at 2 h. Another increase was observed at 3 h for all treatments: 0.2 g/L to 3390 mg/L, 0.6 g/L to 5328 mg/L, and 1 g/L to 7636 mg/L. The control also reported a slight increase to 809 mg/L at 3 h. At 4 h, all treatments except for the control reported an increase in concentration: 0.2 g/L to 4255 mg/L, 0.6 g/L to 6878 mg/L, and 1 g/L to 8356 mg/L. An increase in concentration occurred at 5 h across all treatments: 0.2 g/L to 5003 mg/L, 0.6 g/L to 8065 mg/L, and 1 g/L to 9547 mg/L. The control increased marginally to 926 mg/L at 5 h. There was a decrease in gluconic acid for all treatments at 6 h: control to 723 mg/L, 0.2 g/L to 4912, 0.6 g/L to 6932 mg/L, and 1 g/L to 7968 mg/L. At 7 h, all treatments started to increase in concentration: 0.2 g/L to 5062 mg/L, 0.6 g/L to 7418 mg/L, and 1 g/L to 8521 mg/L. At 8 h, there was an increase in concentration across all treatments: 0.2 g/L to 5344 mg/L, 0.6 g/L to 8238 mg/L, and 1 g/L to 9557 mg/L. The control decreased in concentration at 8 h to 697 mg/L. An increase was observed at 9 h across all treatments: 0.2 g/L to 5538 mg/L, 0.6 g/L to 8784 mg/L, and 1 g/L to 10,081 mg/L. There was a slight increase in concentration for the control at 9 h to 718 mg/L. At 10 h, there was another increase in concentration for all treatments: 0.2 g/L to 6094 mg/L, 0.6 g/L to 9742 mg/L, and 1 g/L to 11,865 mg/L. The control also observed an increase in concentration to 865 mg/L. After 10 h passed, samples were taken at the 24 h mark and all treatments continued to increase: 0.2 g/L to 8818 mg/L, 0.6 g/L to 15,711 mg/L, and 1 g/L increased to 20,485 mg/L. As expected there was a large increase from the initial concentration of gluconic acid over a 24 h period.

It is interesting to note the rate of glucose: gluconic acid production after 24 h of treatment. In the 1 g/L treatment, glucose decreased by 19 g while 20 g of gluconic acid were formed. In the 0.6 g/L treatment, 16 g of glucose were lost and 15 g of gluconic acid were produced, and finally, in the 0.2 g/L treatment, 10 g/L of glucose was lost while 9 g of gluconic acid was formed. This indicates a conversion rate of approximately 1:1 glucose to gluconic acid, similar to previous reports in the literature [4,5].

It was important to observe any color changes throughout the experiment as issues with browning and excessive oxidative effects on treated wines have been previously reported [4,6,7]. The absorption units (AU) means at a 420 nm wavelength were generated for each sample at every time interval and can be found in Table 4.

The initial AU concentration at 0 h for the control was 29, 28 for 0.2 g/L, 26 for 0.6 g/L, and 28 for 1 g/L. There was a significant increase in absorption levels at 1 h for all treatments except for the control: 50 AU for 0.2 g/L, 42 AU for 0.6 g/L, and 39 AU for 1 g/L. At 2 h, there was an increase in absorption for all treatments. The highest absorption was observed in the 0.2 g/L treatment at 56 AU, 53 AU for 0.6 g/L, and 45 AU for 1 g/L. An increase in absorption levels was observed for 0.6 g/L at 53 AU and 1 g/L at 50 AU at 3 h. There was no change in absorption levels for 0.2 g/L and the control at 3 h. At 4 h, there was a decrease in absorption levels for 0.6 g/L at 55 AU. There were no changes in absorption for the other treatments at 4 h. Then at 5 h, all treatments experienced an increase in absorption: 58 AU for 0.2 g/L, 56 AU for 0.6 g/L, and 52 AU for 1 g/L. There was no change in absorption levels for the control at 5 h. There was a steady decline in absorption levels by all treatments from 5 h to 9 h. The absorption levels at 9 h were 21 AU for the control, 45 AU for 0.2 g/L, 39 AU for 0.6 g/L, and 33 AU for 1 g/L. At 10 h, there was an increase in absorption for all treatments: 51 AU for 0.2 g/L, 47 AU for 0.6 g/L, and 45 AU for 1 g/L. After 10 h passed, samples were taken at the 24 h mark and all treatments observed a decrease in absorption: 20 AU for the control, 34 AU for 0.2 g/L, 32 AU for 0.6 g/L, and 38 AU for 1 g/L.

Basic wine parameters (alcohol, pH, and TA) were measured post fermentation and are presented in Table 5.

Alcohol was the highest in the control wines and lowest in the 1 g/L treatment, most likely due to the decrease in glucose after treatment. pH was slightly higher in all treatments at the time of analysis than immediately post treatment, but slightly lower in the control wine. Titratable acidity increased dramatically in the treated wines, with the highest value of 13 g/L in the 1 g/L treatment. From an applied perspective, the 0.6 g/L treatment generated the most balanced results strictly from a chemical point of view (12.7% alcohol, pH of 3.4, and a TA of 11). According to these observations, the ratio of gluconic acid increase (g/L) to TA increase (g/L expressed in tartaric acid) was around 2.5–3:1. Similar results have been reported by Pickering et al. [2].

### 2.2. Sensory Analysis-Flash Profiling

The 10 panelists used 121 aroma attributes and 105 flavor (i.e., in-mouth) attributes to rate the three treated samples and untreated control. The number of attributes generated by each panelist ranged from 2–4 for color, 7–20 for aroma, and 5–19 for flavor. Following removal of redundant attributes, a final total of 23 color attributes, 86 aroma attributes, and 81 flavor attributes were included for analysis. GPA analysis of variance (PANOVA) indicated that the greatest transformation effect for color, aroma, and flavor was for translation (correction for variation associated with attribute intensities [14]. Scale transformation (correction for variations associated with the use of different scale amplitudes by panelists [14], rotation transformation (correction for different interpretations of the terms and indicates the panelists’ agreement or disagreement with respect to the sample [14], and translation were all significant for color (F = 3.13, *p* < 0.01; F = 5.91, *p* < 0.0001; F = 18.66, *p* < 0.0001) and flavor, while only scaling and translation were significant for aroma (F = 2.16, *p* < 0.05; F = 0.77, *p* > 0.05; F = 20.35, *p* < 0.0001). GOX-treated samples were clearly separated from the control (untreated) samples by color (Figure 2A). The first two dimensions explained 98% of the total variance, with the first dimension accounting for 87% of variation with the GOX-treated samples described as yellow-golden, while the untreated control samples were characterized with more orange-red hue attributes.

Over the sample consensus space for both aroma and flavor GPAs (Figure 3A and Figure 4A, respectively), the greatest distance was observed between the untreated control sample (control, with evaluation replications (a, b, c) and the sample that had the highest level of GOX treatment (high GOX), with evaluation replications (a, b, c). The first two dimensions explained 64% of the total variance for aroma (Figure 3) and 73% for flavor (Figure 4).

For both aroma and flavor, dimension 1 explained the greatest variation and clearly differentiated samples with and without GOX treatment (Figure 3B and Figure 4B, respectively). For aroma, dimension 1 separated samples with and without GOX treatment with attributes typically associated with more acidic versus more alcoholic wines, respectively. Similarly for flavor, dimension 1 separated GOX-treated samples as sour, citrus, and floral, while untreated samples were characterized as alcoholic, hot, pungent, bitter, and astringent (Figure 4).

## 3. Discussion

As evidenced by the results, the use of the commercially available mixture of glucose oxidase and catalase Catazyme^®^ 25 L led to significant decreases in pH in a white wine, as hypothesized. At the highest concentration of 1 g/L, the pH decreased from 3.9 to 3.2 in 24 h, while at the lowest, 0.2 g/L, it decreased from 4.0 to 3.5 in the same time interval. This is not only statistically significant, but also significant from an applicability perspective. A short pre-fermentation treatment in the winery is feasible and easy to achieve without any significant cost increases. The shift in pH was both time dependent as well as dose dependent, which means that winemakers can control the treatment either by manipulating the dose or increasing or decreasing the treatment time. In a real-world scenario, given a starting wine pH of 3.9 (not uncommon in warm climates) and a target pH of 3.4 at the highest dose tested here, the reaction time needed could be as short as 10 h. We can also speculate that at higher doses, this time would decrease even further. Furthermore, with our starting pH of 3.9–4.0, we noticed no difficulty in enzyme activation as the enzymatic reactions appeared to start at the time of addition, based on an increase seen in the gluconic acid levels at 1 h (Table 3). From a sensorial perspective, GOX treated wines were clearly distinguishable from the control wines both visually, where control wines were more often described as darker (orange, amber, gold) as opposed to all treated wines (yellow being the dominant descriptor, Figure 1) as well as from an aroma and flavor perspective. The aroma of the control wines was often described as “musty”, “stale”, “oxidized”, or “spoiled” while the treated wines were characterized as “floral”, “fruity”, “sweet”, “candy”, “citrus”, etc. (Figure 2B). Similarly, flavor descriptors for non-treated wines included “bitter”, “chemical”, “tobacco”, “pungent”, and “medicinal” while the aromas of GOX treated wine were dominated by “sour”, “citrus”, “fruit”, and “floral”. The increase in acidity was clearly perceptible and led to the use of descriptors such as “sour” and “acidic” while at the same time increasing the perception of fruitiness, underlined by the use of descriptors such as ”fruit”, “pear”, “citrus”, “apple”, and “grape”, a phenomenon previously described in other research papers [4]. Contrary to our findings, some previous research reported a loss in certain fruit related characteristics (such as lime, for example) [4,11], which was attributed to the aeration process (aeration is necessary during enzymatic treatment as the reaction is oxygen dependent). It is important to note that in some of those studies, the aeration process lasted much longer (up to 72 h), most likely leading to excessive oxidation. We conclude that the high pH of the control wines led to microbiological instability and browning while the increase in pH in the treated wines prevented these processes and led to cleaner, brighter wines.

## 4. Materials and Methods

### 4.1. Enzyme

Catazyme^®^ 25 L was obtained from Novozymes (Bagsvaerd, Denmark) and used for the treatments. The preparation was a liquid mixture of GOX and catalase as per the manufacturer’s data sheet. Treatments were identified as control with no enzyme addition (C) and 0.2 g/L (A), 0.6 g/L (B), and 1.0 g/L (D) of Catazyme^®^ 25 L addition, respectively. Two replicates of each treatment were produced.

### 4.2. Grape Selection and Wine Production

*Riesling* grapes were purchased from Messina Hof Winery in Bryan, TX, USA. *Riesling* grapes were machine harvested (Selective’ Process 2, Pellenc, Notre Dame, France) when the pH reached 4, frozen upon arrival, and kept frozen until ready to use.

### 4.3. Wine Production

Grapes were allowed 48 h to thaw before use. Grapes were pressed using a basket press (45HL Press, Europa, Paso Robles, CA, USA) and juice was collected. Titratable acidity (TA; TA meter, Laboratories Dujardin-Salleron, Noizay, France) and pH (Laboratories Dujardin-Salleron, Noizay, France) were tested when juice was collected.

Ten L of *Riesling* juice were distributed into 6-gallon carboys (6-Gallon Glass Carboy, Northern Brewer, Milwaukee, WI, USA) for each treatment for a total of eight carboys. Environmental air was added at 10 mg/L to the juice for 24 h through a 20–60-gallon aquarium air pump (Tetra, Melle, Germany) to all treatments and replicates. Air flow rate was measured using a digital manometer (R3002, Reed, Wilmington, NC, USA). After the enzyme addition, the glucose oxidase cycle was monitored. pH, d-glucose, d-gluconic acid, and color were measured once an hour during the initial 10 h and again at 24 h using a Gallery™ Automated Photometric Analyzer (98611001,Thermo Fisher Scientific, Vantaa, Finland) and the recommended manufacturer reagents and standards.

After 24 h, aeration was stopped and all treatments were tested for: pH, d-glucose, gluconic acid, and color using the same equipment as above.

Each treatment was then inoculated with yeast nutrient GoFerm (Lallemand, Montreal, QC, Canada) and *Saccharomyces cerevisiae* (Anchor Alchemy I, Scott Laboratories, Petaluma, CA, USA). The wines were fermented to dryness with no difficulty on any of the treatments. Wines were then racked, sulfited with 50 ppm SO_2_ in the form of potassium metabisulfite powder, filtered (Buon Vino Mini Jet Filter, No. 1–3 pads), and labeled, according to treatment. Wine was stored at 5 °C until use. Wines were analyzed for alcohol concentration using an Anton Paar alcoholizer (Anton Paar, Ashland, VA, USA), TA, and pH using a Dujardin-Salleron automatic titralyzer (Dujardin-Salleron, Noizay, France). Sensory evaluation was performed after four months of storage.

### 4.4. Sensory Evaluation 

Flash profiling was conducted on the control and GOX-treated wines [14,15,16,17,18]. The rationale and utility of flash profiling are discussed in detail elsewhere [14,16,17,18]. The method used here was based on Bredie et al. (2018) [15]. Evaluation sessions were conducted in the sensory analysis laboratory at Texas A&M University. The panel was composed of 10 assessors (five males and five females aged 21–65 years) with over 200 h of training and experience in descriptive analysis.

Samples (30 mL) were taken out of a cooler and served cold (10 °C) in 12 oz wine glasses labeled with three-digit codes. The presentation of the samples was randomized for each panelist. The first session consisted of attribute generation, followed by three days of testing.

The panel was asked to identify aroma, flavor, and color attributes from four treated Riesling wines with each treatment having two replicates (C1, C2, A1, A2, B1, B2, D1, D2). The first session was conducted in an air-controlled room (24 °C) for attribute development and sample analysis. The total session time was 2 h. All eight samples were served all at once and panelists were given 1 h to generate attributes. A 30-min break was given to panelists, while attributes were collated and written down on a whiteboard by the panel leader. Panelists were then asked to observe the total attributes accumulated and instructed to add or subtract attributes to their own list as they felt appropriate. Individual attributes were finalized and recorded for testing purposes.

The second, third, and fourth days consisted of testing sessions. Testing sheets were made for each panelist based on the final attribute list they had generated the previous day. Instructions were clearly stated on the testing sheets. Panelists were asked to rank attributes according to intensity on an ordinal scale anchored from ‘low’ to ‘high’. In order, panelists evaluated aroma attributes, flavor attributes, and then color attributes, taking a 30 min break between each section. All eight samples (30 mL) were placed in temperature controlled (24 °C) individual booths and presented under red lights. Red lights were used for aroma and flavor evaluations. Upon completion of the aroma and flavor sections, lights were changed from red to white for color evaluation. As panelists evaluated each wine, they were asked to cleanse their palates using distilled water and unsalted crackers.

#### Data Analysis

The aroma, flavor, and color data were analyzed using generalized procrustes analysis (GPA; Gower, 1975) using the Commandeur method in XLSTAT (v2018, Addinsoft, New York, NY, USA). A predetermined alpha of 0.05 was selected. The flash profile sensory data were collected in triplicate for each replication. Each replication was averaged for each panelist and each attribute before the GPAs.

The chemical data were analyzed using ANOVA in JMP (v14.0, SAS Institute, Cary, NC, USA) and XLSTAT (v2020, Addinsoft, New York, NY, USA) with a predetermined alpha of 0.05. Treatment and rep were used as fixed effects. When the F-test was determined to be significant, the Student’s *t*-test was utilized for post-hoc comparison of means. For the finished wine data, Tukey’s HSD was used for separation of the means.

## 5. Conclusions

New technologies are needed to improve high pH wine quality, as it often can lead to microbial instability, oxidation, unpleasant aroma, and flavor profiles and color stability issues. This issue is particularly pervasive in warm climate regions such as Texas, where ripening balance at the time of harvest is often very hard to achieve due to the dramatic drop in acidity levels before full phenolic ripeness is reached. Catazyme^®^ 25 L (glucose oxidase with catalase) was used to enzymatically metabolize glucose into gluconic acid, leading to an increase in total acidity. The pH was decreased in all treatments over a 24 h period, with a maximum decrease of 0.7 pH in the 1 g/L treatment. A flash sensory profile method was implemented to evaluate the flavor, aroma, and color of the wines. Treatments with Catazyme^®^ 25 L led to an increase in positive color, aroma, and flavor attributes, whereas the ones without were described using more negative attributes. The control wine without Catazyme^®^ 25 L had clear signs of oxidation and color changes, most likely attributable to the high pH. At the beginning of the experiment, the control was lighter in appearance while the treatments were dark, most likely due to hyperoxidation in the presence of hydrogen peroxide and air of the treated wines. The final wine color was the opposite of the initial observations. After filtration and SO_2_ addition, the quality of the treatment wines (with Catazyme^®^ 25 L) was preserved and reflected more positive sensory attributes during sensory analysis, although some indication of increased volatile acidity was also noted, possibly due to the presence of high concentrations of gluconic acid (volatile acidity was not tested in this study). The control wine was dark brownish yellow in appearance and was found to have negative sensory attributes during sensory analysis. Treatment of grape juice with Catazyme^®^ 25 L led to a significant decrease in the pH of Riesling juice, resulting in increased wine quality, as hypothesized. Real-life applications of this method need not be as extreme as this study, as the enzymatic reaction could be stopped when reaching a target pH (most likely less than 24 h reaction time), thus leading to lower gluconic acid levels, lessening its impact on the sensory profile of the wines.

It is important to note that both GOX and catalase are GRAS (generally regarded as safe) by the FDA (Food and Drug Administration) in the U.S. and are permitted for use in wine [19]. The OIV (International Organization of Vine and Wine) does not specifically regulate the use of the two enzymes, although enzyme preparations in general are permitted for use in grape juice and wine as long as they are safe from a health perspective and do not negatively affect the quality of the wines [20].

Possible further research topics could include the applicability of this method for red wines, application timing (for example at the time of yeast inoculation), application to a target pH, and SO_2_ behavior over time.

## Figures and Tables

**Figure 1 molecules-26-02730-f001:**
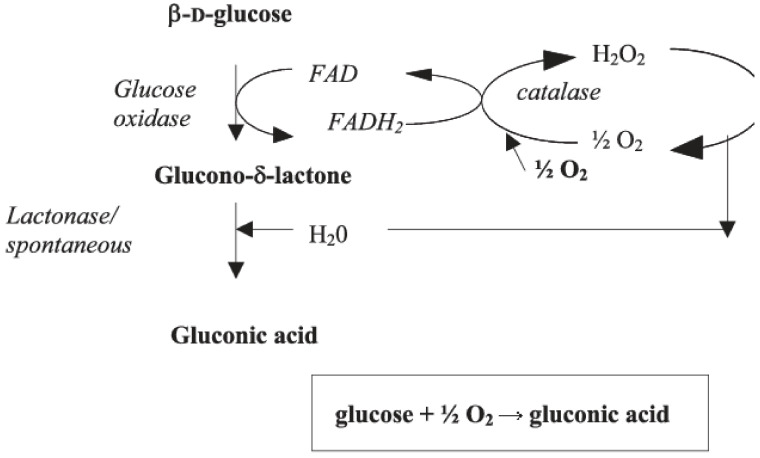
Oxidation of glucose by glucose oxidase (Reprinted from Ref. [8]).

**Figure 2 molecules-26-02730-f002:**
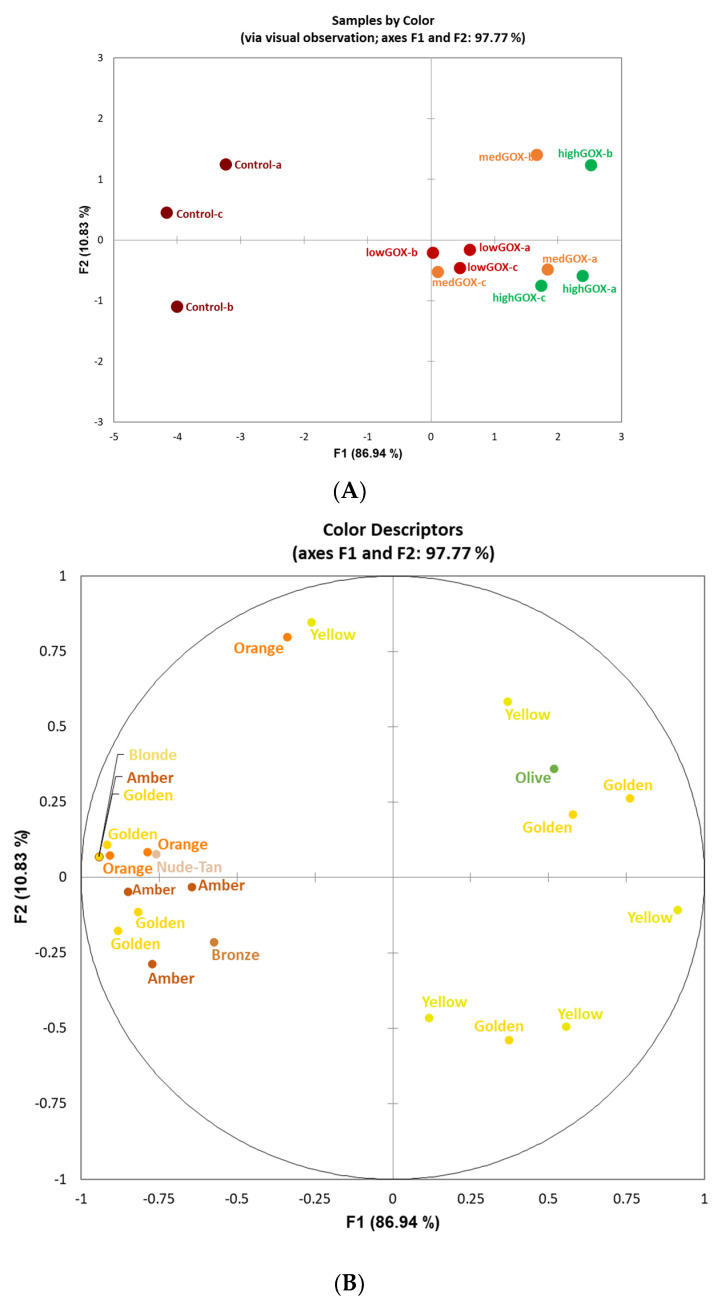
Plots of (**A**) sample consensus space by (**B**) color attributes (**A**). Sample consensus space of untreated (control) samples and GOX-treated samples by level (lowGOX, medGOX, highGOX) and evaluation replicate (a, b, c) resulting from (**B**) color characterization by panelists using flash profile via generalized procrustes analysis. Attribute words are colored to match.

**Figure 3 molecules-26-02730-f003:**
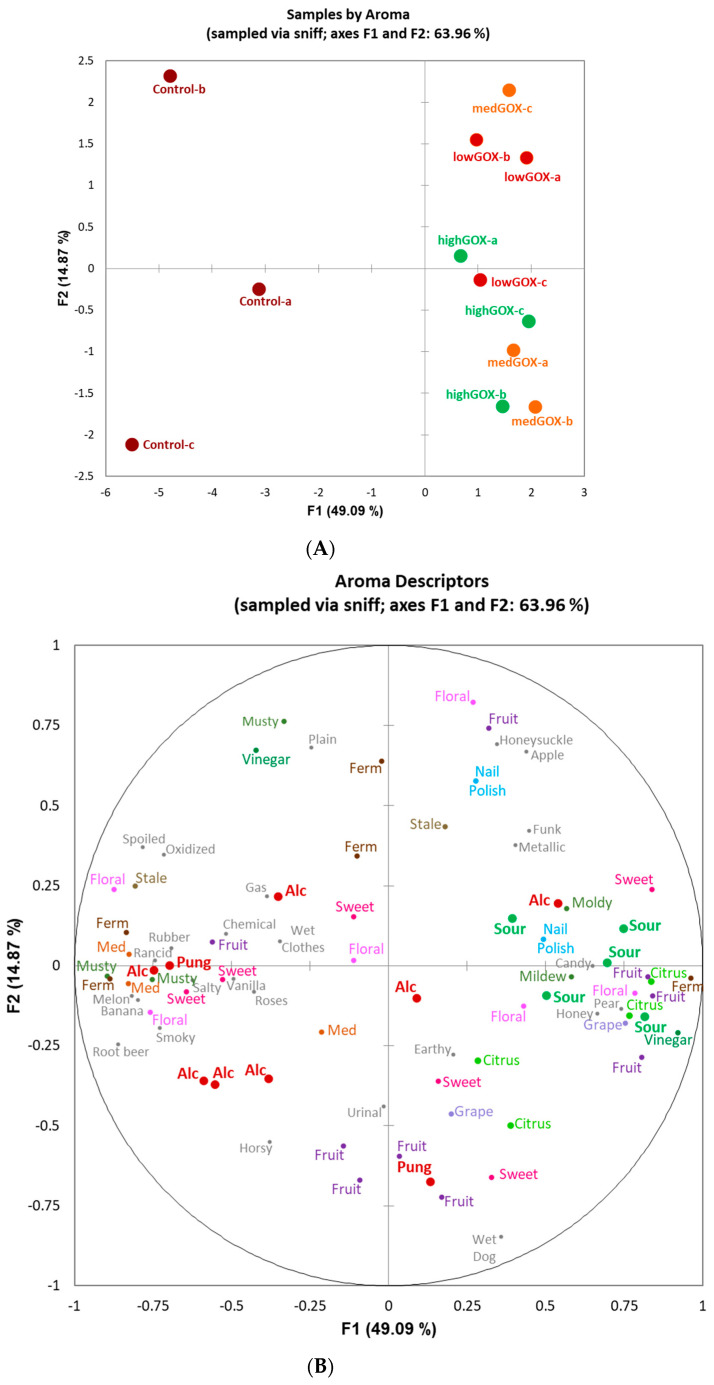
Plots of (**A**) sample consensus space by (**B**) aroma (via sniff) attributes (**A**). Sample consensus space of the untreated (control) samples and GOX-treated samples by level (lowGOX, medGOX, highGOX) and evaluation replicate (a, b, c) resulting from (**B**) aroma characterization by panelists using flash profile via generalized procrustes analysis. Similar colors of attribute words indicate similar aroma notes. Alc = Alcoholic, Ferm = Fermented, Med = Medicinal, Pung = Pungent.

**Figure 4 molecules-26-02730-f004:**
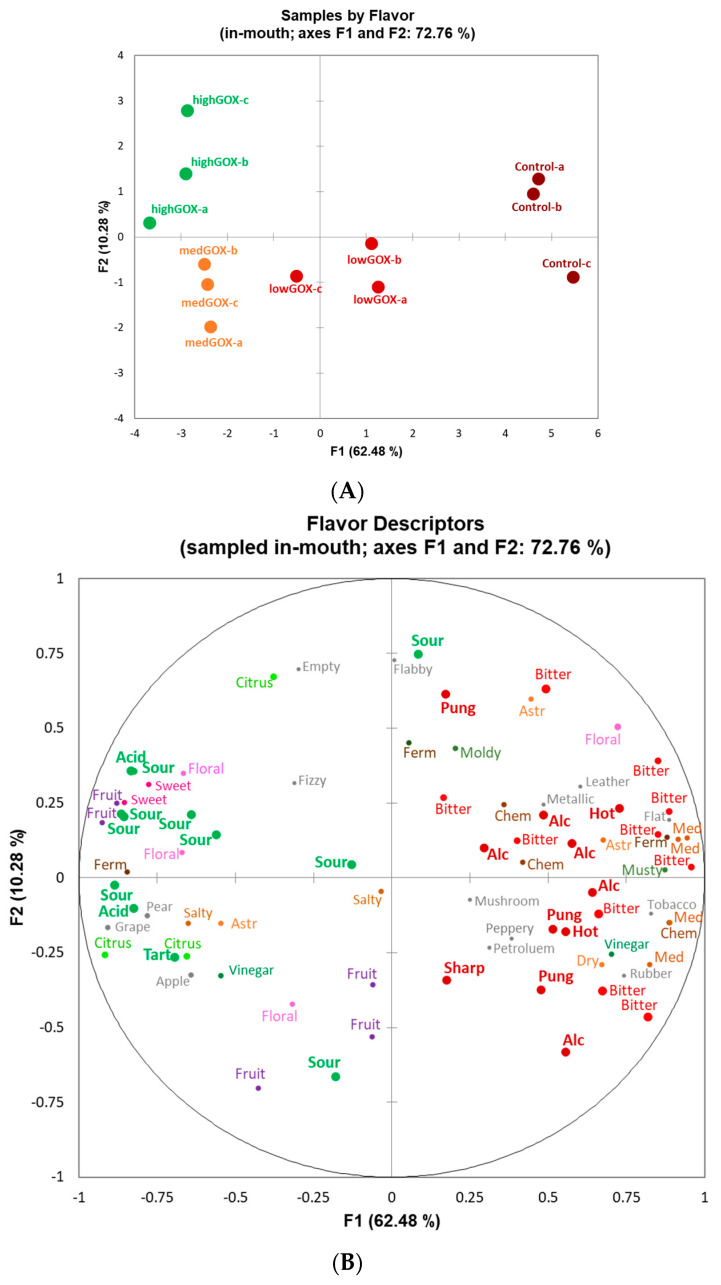
Plots of (**A**) sample consensus space by (**B**) flavor (sampled in-mouth) attributes (**A**). Sample consensus space of untreated (control) samples and GOX-treated samples by level (lowGOX, medGOX, highGOX) and evaluation replicate (a, b, c) resulting from (**B**) flavor characterization by panelists using flash profile via generalized procrustes analysis. Similar colors of attribute words indicate similar flavor notes. Acid = Acidic, Alc = Alcoholic, Chem = Chemical, Ferm = Fermented, Med = Medicinal, Pung = Pungent.

**Table 1 molecules-26-02730-t001:** Initial pH values over the first 24 h.

		Treatment			
Hour	*p*-Value ^e^	Control	0.2 g/L	0.6 g/L	1 g/L
0	0.81	4.0 ± 0.08	4.0 ± 0.01	3.9 ± 0.01	3.9 ± 0.03
1	0.11	3.9 ± 0.00	3.9 ± 0.01	3.9 ± 0.00	3.9 ± 0.00
2	0.03	4.0 ± 0.03 ^a^	3.9 ± 0.02 ^ab^	3.8 ± 0.01 ^b^	3.8 ± 0.01 ^b^
3	0.01	3.9 ± 0.00 ^a^	3.8 ± 0.01 ^ab^	3.7 ± 0.01 ^bc^	3.7 ± 0.01 ^c^
4	0.009	3.9 ± 0.00 ^a^	3.8 ± 0.00 ^b^	3.7 ± 0.01 ^c^	3.7 ± 0.00 ^c^
5	0.002	3.9 ± 0.01 ^a^	3.7 ± 0.00 ^b^	3.6 ± 0.01 ^c^	3.6 ± 0.00 ^c^
6	0.002	3.9 ± 0.01 ^a^	3.7 ± 0.00 ^b^	3.6 ± 0.01 ^c^	3.6 ± 0.00 ^c^
7	0.001	3.9 ± 0.01 ^a^	3.7 ± 0.00 ^b^	3.6 ± 0.00 ^c^	3.5 ± 0.00 ^c^
8	0.0005	3.9 ± 0.01 ^a^	3.6 ± 0.01 ^b^	3.5 ± 0.01 ^c^	3.5 ± 0.00 ^c^
9	0.0003	3.9 ± 0.01 ^a^	3.6 ± 0.00 ^b^	3.5 ± 0.00 ^c^	3.5 ± 0.00 ^c^
10	0.001	3.9 ± 0.01 ^a^	3.6 ± 0.01 ^b^	3.5 ± 0.01 ^c^	3.5 ± 0.00 ^c^
24	<0.0001	3.9 ± 0.01 ^a^	3.5 ± 0.00 ^b^	3.3 ± 0.00 ^c^	3.2 ± 0.00 ^d^

^abcd^ Mean values within a row followed by the same letter were not significantly different (*p* > 0.05). ^e^ *p*-value from analysis of variance tables. 0.2 g/L (low), 0.6 g/L (medium), and high (1 g/L) indicate the rate of Catazyme^®^ 25 L addition to treated wines.

**Table 2 molecules-26-02730-t002:** Initial glucose (g/L) values over the first 24 h.

Hour	*p*-Value ^a^	Control	0.2 g/L (Low)	0.6 g/L (Medium)	1 g/L (High)
0	0.72	108 ± 0.3	110 ± 0.3	111 ± 0.4	110 ± 0.6
1	0.76	108 ± 0.6	109 ± 0.3	110 ± 0.8	107 ± 0.6
2	0.86	109 ± 0.3	111 ± 0.5	110 ± 0.4	109 ± 0.2
3	0.42	110 ± 0.6	112 ± 0.7	112 ± 0.7	109 ± 0.2
4	0.47	113 ± 0.6	111 ± 0.6	110 ± 0.4	107 ± 0.2
5	0.54	113 ± 0.6	113 ± 0.5	112 ± 0.3	108 ± 0.6
6	0.67	105± 0.6	105 ± 0.8	103 ± 0.6	102 ± 0.3
7	0.34	106 ± 1.1	105 ± 1.0	103 ± 1.1	101 ± 1.0
8	0.2	107 ± 0.9	105 ± 0.4	101 ± 0.6	100 ± 1.5
9	0.3	105 ± 0.2	104 ± 0.3	102 ± 0.4	99 ± 0.3
10	0.13	107 ± 0.3	104 ± 0.8	101 ± 0.7	94 ± 0.9
24	0.09	103 ± 0.6	100 ± 0.3	95 ± 0.3	91 ± 0.4

^a^ *p*-value from analysis of variance tables. 0.2 g/L, 0.6 g/L, and 1 g/L indicate the rate of Catazyme^®^ 25 L addition to treated wines.

**Table 3 molecules-26-02730-t003:** Initial gluconic acid (mg/L) values over the first 24 h.

		Treatment			
Hour	*p*-Value ^e^	Control	0.2 g/L (Low)	0.6 g/L (Medium)	1 g/L (High)
0	0.25	627 ± 10.0	346 ± 18.9	263 ± 11.0	327 ± 35.8
1	0.008	682 ± 21.1 ^c^	1274 ± 11.3 ^c^	2126 ± 55.8 ^b^	3153 ± 62.6 ^a^
2	0.0008	709 ± 23.1 ^d^	2259 ± 35.5 ^c^	3717 ± 21.9 ^b^	4810 ± 43.4 ^a^
3	0.0002	809 ± 24.7 ^d^	3391 ± 15.8 ^c^	5328 ± 23.6 ^b^	7636 ± 101.9 ^a^
4	0.001	864 ± 7.3 ^d^	4255 ± 28.1 ^c^	6878 ± 60.8 ^b^	8356 ± 38.0 ^a^
5	0.0001	925 ± 29.6 ^d^	5003 ± 15.3 ^c^	8065 ± 189.7 ^b^	9547 ± 75.7 ^a^
6	<0.0001	723 ± 13.9 ^d^	4912 ± 35.8 ^c^	6932 ± 99.8 ^b^	7968 ± 56.4 ^a^
7	0.0004	742 ± 30.4 ^d^	5062 ± 85.2 ^c^	7418 ± 335.6 ^b^	8521 ± 296.8 ^a^
8	0.0004	697 ± 33.7 ^d^	5344 ± 47.6 ^c^	8238 ± 288.4 ^b^	9557 ± 289.8 ^a^
9	<0.0001	718 ± 18.7 ^d^	5538 ± 47.0 ^c^	8784 ± 72.8 ^b^	10,081 ± 67.5 ^a^
10	0.0004	865 ± 7.9 ^d^	6094 ± 32.9 ^c^	9742 ± 48.7 ^b^	11,865 ± 59.9 ^a^
24	0.0001	846 ± 20.5 ^d^	8818 ± 75.2 ^c^	15,711 ± 92.0 ^b^	20,485 ± 112.1 ^a^

^abcd^ Mean values within a row followed by the same letter were not significantly different (*p* > 0.05). ^e^
*p*-value from analysis of variance tables. 0.2 g/L, 0.6 g/L, and 1 g/L indicate the rate of Catazyme^®^ 25 L addition to treated wines.

**Table 4 molecules-26-02730-t004:** Initial wine CL (AU at 420 nm) values over the first 24 h.

		Treatment			
Hour	*p*-Value ^d^	Control	0.2 g/L (Low)	0.6 g/L (Medium)	1 g/L (High)
0	0.88	29 ± 0.2	28 ± 0.3	26 ± 0.2	28 ± 0.1
1	0.007	29 ± 0.2 ^b^	50 ± 0.5 ^a^	42 ± 0.7 ^ab^	39 ± 0.1 ^ab^
2	0.03	29 ± 0.1 ^b^	56 ± 0.3 ^a^	53 ± 0.2 ^a^	45 ± 0.3 ^a^
3	0.01	30 ± 0.3 ^b^	56 ± 0.2 ^a^	57 ± 0.4 ^a^	50 ± 0.3 ^a^
4	0.004	29 ± 0.3 ^b^	55 ± 0.2 ^a^	55 ± 0.2 ^a^	50 ± 0.5 ^a^
5	0.007	28 ± 0.2 ^b^	58 ± 0.6 ^a^	56 ± 0.2 ^a^	52 ± 0.4 ^a^
6	0.01	24 ± 0.2 ^b^	54 ± 0.1 ^a^	50 ± 0.3 ^a^	46 ± 0.3 ^a^
7	0.008	24 ± 0.2 ^c^	51 ± 1.8 ^a^	49 ± 0.2 ^ab^	41 ± 0.3 ^b^
8	0.008	22 ± 0.1 ^b^	53 ± 0.2 ^a^	48 ± 0.1 ^a^	42 ± 0.4 ^a^
9	0.006	21 ± 0.1 ^c^	45 ± 0.1 ^a^	39 ± 0.1 ^ab^	33 ± 0.2 ^b^
10	0.01	22 ± 0.1 ^b^	51 ± 0.5 ^a^	47 ± 0.2 ^a^	45 ± 1.3 ^a^
24	0.3	20 ± 0.1	34 ± 0.2	32 ± 0.1	38 ± 0.3

^abc^ Mean values within a row followed by the same letter were not significantly different (*p* > 0.05). ^d^ *p*-value from analysis of variance tables. 0.2 g/L, 0.6 g/L, and 1 g/L indicate the rate of Catazyme^®^ 25 L addition to treated wines.

**Table 5 molecules-26-02730-t005:** Basic wine parameters post fermentation. Each treatment rep was analyzed in duplicate. Results presented are averages over treatment replicates. ^abcd^ Mean values within a column followed by the same letter were not significantly different (*p* > 0.05). 0.2 g/L, 0.6 g/L, and 1 g/L indicate the rate of Catazyme^®^ 25 L addition to treated wines.

	Alcohol (% by vol.)	pH	TA (g/L)
**Control**	13.5 ± 0.09 ^a^	3.9 ± 0.04 ^a^	5 ± 0.01 ^d^
**Low (0.2 g/L)**	13.0 ± 0.07 ^b^	3.6 ± 0.07 ^b^	9 ± 0.04 ^c^
**Medium (0.6 g/L)**	12.7 ± 0.04 ^b^	3.4 ± 0.04 ^c^	11 ± 0.04 ^b^
**High (1 g/L)**	12.4 ± 0.04 ^c^	3.3 ± 0.07 ^d^	13 ± 0.04 ^a^

## Data Availability

The data presented in this study are openly available at the Texas Data Repository at [https://doi.org/10.18738/T8/3GPWGF, accessed on 16 March 2021].

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
