# Peer review of "Enzymatic Management of pH in White Wines"

_molecules, 2021, doi:10.3390/molecules26092730_

Round 1

Reviewer 1 Report

The manuscript is interesting and easy to follow. However, there are some issues that need to be addressed by the authors before it is accepted for publication.

  • Please use litters instead of gallons.
  • Please provide more details about the sensory analysis.
  • Please provide more details about the statistical analysis, including validation of the models.

Author Response

The manuscript is interesting and easy to follow. However, there are some issues that need to be addressed by the authors before it is accepted for publication.

  • Please use litters instead of gallons.
    • This change has been made throughout.
  • Please provide more details about the sensory analysis.
    • The references noted below have been added as background on the rationale for using the Flash Profile methodology overall and specifically in wine sensory evaluation. Please let us know if there are specific details you are requesting be added.

Liu et al. Performance of Flash Profile and Napping with and without training for describing small sensory differences in a model wine. Food Quality and Preference. Volume 48, Part A, March 2016, Pages 41-49.

Liu et al. Comparison of rapid descriptive sensory methodologies: Free-Choice Profiling, Flash Profile and modified Flash Profile. Food Research International. Volume 106, April 2018, Pages 892-900.

Škrobot et al. Flash profile as a rapid descriptive analysis in sensory characterization of traditional dry fermented sausages. Food and Feed Research. Volume 47, 2020. Part 1, 55-63.

  • Please provide more details about the statistical analysis, including validation of the models.
    • We would be happy to add more details but are unclear about the validation requested by the reviewer. The current work is not intended to be a methods paper or provide methodological comparisons, where we agree that validation would be required. Otherwise, validation of the findings presented here would require conducting the profiling again to compare the results, which we believe is outside of the scope of the current research.  

Reviewer 2 Report

The study appears interesting as it deals with wine issues due to temperature increases and climatic changes.

My concerns are regard line 98-104 where the authors stated that, as reported in the table, 2 D glucose changes its concentration over the 24 hours and, even, that  ‘A change in concentration for all treatments happened at 6h. The biggest decrease occurred for 1 g/L with a concentration   of 101.5 g/L, 0.6 g/L at 103.3 g/L, 0.2 g/L to 105 g/L, and control to 105 g/L’

These changes are not supported by statistical analysis significance so, any changes should not be highlighted and discussed.  

In case changes were meant as vertical comparison over the hours these changes should be supported by statistical analysis as 'repeated measures' that should be provided.

Moreover, given that it is quite strange that at hour 10  horizontal values  94.2 (1 g /L) against 106.8 (control)  are not statistically different it should be very crucial that each mean data was followed by standard deviation besides letters. 

This method should be utilized for each table shown.  

Mostly, the described approach above should be adopted in table 5 where comparisons in Alcohol content (% by vol) are showed in relation to different treatments without any statistical confirmation. 

Without this information, any conclusion cannot be assumed as reliable. 

'Materials and Methods' should be moved and put after the 'Introduction'.

Finally, as a minimum, given that the title deals with climatic influence on wines characteristics, it should be provided with a monthly climatic description of the area where the analyzed wines come for.   In case of lack of climatic parameters of the year of the trial and of a description of the region any climatic reference should not be reported.

To conclude, this kind of research should be confirmed by, at least, another year of trial.

Author Response

Thank you for your comments and suggestions!

We are addressing them, individually, below:

"My concerns are regard line 98-104 where the authors stated that, as reported in the table, 2 D glucose changes its concentration over the 24 hours and, even, that  ‘A change in concentration for all treatments happened at 6h. The biggest decrease occurred for 1 g/L with a concentration   of 101.5 g/L, 0.6 g/L at 103.3 g/L, 0.2 g/L to 105 g/L, and control to 105 g/L’

These changes are not supported by statistical analysis significance so, any changes should not be highlighted and discussed.  

In case changes were meant as vertical comparison over the hours these changes should be supported by statistical analysis as 'repeated measures' that should be provided.

Moreover, given that it is quite strange that at hour 10  horizontal values  94.2 (1 g /L) against 106.8 (control)  are not statistically different it should be very crucial that each mean data was followed by standard deviation besides letters."

Noted. We reduced the paragraph on glucose and eliminated the term "decreased". We added a comment that a trend for lower values can be seen and added the standard deviations throughout all tables, as suggested

"Mostly, the described approach above should be adopted in table 5 where comparisons in Alcohol content (% by vol) are showed in relation to different treatments without any statistical confirmation. 

Without this information, any conclusion cannot be assumed as reliable. "

We ran statistical analysis (ANOVA followed by Tukey's HSD) on the data and added the information in the table (Table 5).

"'Materials and Methods' should be moved and put after the 'Introduction'."

We followed the instructions given by the Journal to authors, found here https://www.mdpi.com/journal/molecules/instructions#manuscript

Research Manuscript Sections

  • Introduction: The introduction should briefly place the study in a broad context and highlight why it is important. It should define the purpose of the work and its significance, including specific hypotheses being tested. The current state of the research field should be reviewed carefully and key publications cited. Please highlight controversial and diverging hypotheses when necessary. Finally, briefly mention the main aim of the work and highlight the main conclusions. Keep the introduction comprehensible to scientists working outside the topic of the paper.
  • Results: Provide a concise and precise description of the experimental results, their interpretation as well as the experimental conclusions that can be drawn.
  • Discussion: Authors should discuss the results and how they can be interpreted in perspective of previous studies and of the working hypotheses. The findings and their implications should be discussed in the broadest context possible and limitations of the work highlighted. Future research directions may also be mentioned. This section may be combined with Results.
  • Materials and Methods: They should be described with sufficient detail to allow others to replicate and build on published results. New methods and protocols should be described in detail while well-established methods can be briefly described and appropriately cited. Give the name and version of any software used and make clear whether computer code used is available. Include any pre-registration codes.
  • Conclusions: This section is not mandatory, but can be added to the manuscript if the discussion is unusually long or complex.
  • Patents: This section is not mandatory, but may be added if there are patents resulting from the work reported in this manuscript."

"Finally, as a minimum, given that the title deals with climatic influence on wines characteristics, it should be provided with a monthly climatic description of the area where the analyzed wines come for.   In case of lack of climatic parameters of the year of the trial and of a description of the region any climatic reference should not be reported."

Thank you, agreed. We removed Hot Climate from the title, as the technique is applicable to all wines, regardless of source and origin.

"To conclude, this kind of research should be confirmed by, at least, another year of trial."

Thank you. We maintain the view that one year of trial, for enological focused experiments (as opposed to viticulture ones, which traditionally require two or more years of trials) is satisfactory. All other papers referenced here and previously published are based on one year data. 

As a comment, all data, with the exception of glucose, shows significant differences between treatments and control. pH is significantly lower in treated wines, which was the main focus of the paper. Furthermore, gluconic acid significantly increases in treated wines, which clearly indicates enzymatic activity, even in the absence of significance in glucose. TA also significantly increases and alcohol is lower in treated wines confirming a loss in glucose. We feel we are offering strong evidence to support our hypotheses and conclusions.

Reviewer 3 Report

see attached 

Author Response

Thank you for your comments and suggestions!

We are addressing them individually, below:

"...chemical analysis was limited to a few parameters and, at least, acetic acid levels (due the presence of gluconic acid), as well as ethyl acetate, should have been monitored to fully appreciate the outcome of this practice. To confirm my hypothesis some of the descriptors used by the panel are sour, vinegar, oxidised, spoiled, and nail polish (Figures 3 & 4). These all are markers of oxidation/spoilage/acid acetic formation. This is the main issue why GOX does not have a real market among the winemakers looking at using this enzyme. To me, this research should be more presented as a communication, rather than a research article."

Thank you for your insights!

In reviewing the pertinent literature, we have concluded that this practice (adding GOX to juice or must) has no impact on acetic acid production. In the studies that did look at acetic acid concentrations in wines produced with GOX (Pickering 1999, van Rensburg 2009 and Rocker 2016) acetic acid levels were lower in GOX treated wines than in control. There was only one instance (Rocker, 2016) for the large scale GOX treated wine had a slightly higher acetic acid concentration (0.63 g/L for GOX treated wine as opposed to 0.56 g/L for Control wine), but that was not statistically significant.

Thus, we concluded that GOX does not affect acetic acid levels in wines, and decided not to monitor it.

We are not sure to interpret the "acetic acid levels (due the presence of gluconic acid)" comment, as we are not aware of a correlation between the two. 

We concur with the statement that some of the descriptors used (vinegar, oxidised, spoiled, and nail polish) are usually associated with some degree spoilage. We think that the extreme lowering of pH leading to high gluconic acid levels, may generate the potentially unpleasant characters in the wine. However, in practice, there is no need for such extreme steps and the reaction can be easily stopped after a few hours, when pH reaches a target value.

"This is the main issue why GOX does not have a real market among the winemakers looking at using this enzyme."

We are not aware of this but would very much appreciate any source for this statement as it would allow us to address these concerns in future research. From our standpoint this is a very easy to apply technique which is fast, inexpensive and effective.

"L284-287: why air was supplemented to the juice?"

Thank you.

As described in lines 35-38 "The conversion of d-glucose into gluconic acid immobilizes its ability to act as a fermentable sugar and increases acidity [5]. The rate of conversion is primarily dependent on oxygen aeration [2]. The reaction mechanism for the GOX mediated Glucose to Gluconic acid pathways are shown in Figure 1."

The enzymatic reaction is oxygen dependent, thus the need for aeration.

"L294-296: GoFerm and Alchemy are Lallemand owned products? The authors should list the manufacturer not the supplier (Enartis and Scott Labs)."

Thank you. We made the changes, as suggested.

"L302: the authors should be more specific when they describe their chemical/sensory analyses. When was flash profiling performed? How old was the wine by then?"

Thank you. We added " Sensory analysis was performed after four months of storage." in the manuscript.

Conclusions should be re-written. It is not a real conclusion rather a summary of major findings which appear to be redundant. The authors should outline major outcomes for winegrowers, research limitations and further opportunities.

Thank you!

Line 375-377 We added "although some indication of increased volatile acidity was also noted, possibly due to the presence of high concentrations of gluconic acid (volatile acidity was not tested in this study). "

Line 380-386 We added "Real-life applications of this method need not be as extreme as this study, as the enzymatic reaction could be stopped when reaching a target pH (most likely less than 24 hrs reaction time), thus leading to lower gluconic acid levels, lessening its impact on the sensory profile of the wines. Possible further research topics could include the applicability of this method for red wines, application timing (for example at the time of yeast inoculation), application to a target pH and SO2 behavior over time."

Tables 2-4: Please, mention to what: 0.2, 0.6 and 1 g/L, refer using a footnote in the table. 

Thank you. Done for the tables 2-5.

Round 2

Reviewer 1 Report

Accept

Author Response

Thank you.

Reviewer 2 Report

The article has been  ameliorated in the form, it still needs additional experiments for having  scientific confirrmation. It should be useful  to add another year of experimentation before  publishing the data.  

Author Response

Thank you.

We have a new year of data on red wine that confirm our original findings (the ones we are publishing here). We intend to publish that in a separate manuscript.

Reviewer 3 Report

The manuscript has been improved and is ready for publication, at least in my opinion. Just a few extra things.

"L284-287: why air was supplemented to the juice?" (from my previous comment). Sorry, I was actually meaning if some information could have be added to make this clear to the reader.

We are not sure to interpret the "acetic acid levels (due the presence of gluconic acid)" comment, as we are not aware of a correlation between the two.

This is very much dependent on grapes health status/microorganisms/aerobic conditions - please, see pg. 5193 (Altered Carbohydrate Metabolism) in dx.doi.org/10.1021/jf400641r | J. Agric. Food Chem. 2013, 61, 5189−5206. It is available online and should contain more refs. of interest. 

All these descriptors associated with oxidation relate to acetic acid or ethyl acetate. If acetic acid is low, then they should relate to an excess of ethyl acetate (>500 mg/L). 

Author Response

"L284-287: why air was supplemented to the juice?" (from my previous comment). Sorry, I was actually meaning if some information could have be added to make this clear to the reader.

Thank you. We added "(aeration is necessary during enzymatic treatment as the reaction is oxygen dependent)." Lines 276-277

This is very much dependent on grapes health status/microorganisms/aerobic conditions - please, see pg. 5193 (Altered Carbohydrate Metabolism) in dx.doi.org/10.1021/jf400641r | J. Agric. Food Chem. 2013, 61, 5189−5206. It is available online and should contain more refs. of interest. 

Thank you very much for this resource! We acknowledge the fact that microorganisms associated with grape disease can cause the formation of gluconic acid and acetic acid, leading to off aromas. Furthermore, gluconic acid can be sometimes metabolized into acetic acid, but only in the presence of certain microorganisms. In our case, the presence of gluconic acid in wine is not related to microbial activity, but rather to the introduction of a singular purified enzyme (glucose oxidase) that on its own will not continue the metabolic process of gluconic acid to acetic acid.

With that being said, we concur that some of the descriptors mentioned can be associated with elevated volatile acidity and in the future we will monitor VA (acetic acid and ethyl acetate) levels in our research wines.